# Wearable sensor-based performance status assessment in cancer: A pilot multicenter study from the Alliance for Clinical Trials in Oncology (A19_Pilot2)

William A. Wood[1]*, Deepika Dilip[2], Andriy Derkach[2], Natalie S. Grover[1], Olivier Elemento[3], Ross Levine[2], Gita Thanarajasingam[4], John A. Batsis[5], Charlotte Bailey[1], Arun Kannappan[6], Steven M. Devine[7], Andrew S. Artz[8], Jennifer A. Ligibel[9], Ethan Basch[1], Erin Kent[10], Jacob Glass[2]

1 Lineberger Comprehensive Cancer Center and School of Medicine, University of North Carolina, Chapel Hill, North Carolina, United States of America, 2 Memorial Sloan Kettering Cancer Center, New York, New York, United States of America, 3 Department of Physiology and Biophysics, Weill Cornell Medicine, New York, New York, United States of America, 4 Division of Hematology, Mayo Clinic, Rochester, Minnesota, United States of America, 5 School of Medicine and Gillings School of Global Public Health, University of North Carolina, Chapel Hill, North Carolina, United States of America, 6 Division of Pulmonary Sciences and Critical Care Medicine, University of Colorado, Aurora, Colorado, United States of America, 7 National Marrow Donor Program (NMDP)/Be The Match, Minneapolis, Minnesota, United States of America, 8 Department of Hematology and Hematopoietic Cell Transplantation, City of Hope, Duarte, California, 9 Dana Farber Cancer Institute, Boston, Massachusetts, United States of America, 10 Lineberger Comprehensive Cancer Center and Gillings School of Global Public Health, University of North Carolina, Chapel Hill, North Carolina, United States of America

* william_wood@med.unc.edu

**Data Availability Statement:** The data used in this submission can be accessed at the following link: https://github.com/JLGlass/fitness_analysis_code.

## Abstract

Clinical performance status is designed to be a measure of overall health, reflecting a patient's physiological reserve and ability to tolerate various forms of therapy. Currently, it is measured by a combination of subjective clinician assessment and patient-reported exercise tolerance in the context of daily living activities. In this study, we assess the feasibility of combining objective data sources and patient-generated health data (PGHD) to improve the accuracy of performance status assessment during routine cancer care. Patients undergoing routine chemotherapy for solid tumors, routine chemotherapy for hematologic malignancies, or hematopoietic stem cell transplant (HCT) at one of four sites in a cancer clinical trials cooperative group were consented to a six-week prospective observational clinical trial (NCT02786628). Baseline data acquisition included cardiopulmonary exercise testing (CPET) and a six-minute walk test (6MWT). Weekly PGHD included patient-reported physical function and symptom burden. Continuous data capture included use of a Fitbit Charge HR (sensor). Baseline CPET and 6MWT could only be obtained in 68% of study patients, suggesting low feasibility during routine cancer treatment. In contrast, 84% of patients had usable fitness tracker data, 93% completed baseline patient-reported surveys, and overall, 73% of patients had overlapping sensor and survey data that could be used for modeling. A linear model with repeated measures was constructed to predict the patient-reported physical function. Sensor-derived daily activity, sensor-derived median heart rate, and patient-

**Funding:** Research reported in this publication was supported by the National Cancer Institute of the National Institutes of Health under the Award Number UG1CA189823 (Alliance for Clinical Trials in Oncology NCORP Grant), UG1CA233290, UG1CA233324, UG1CA233373, UG1CA232760 and 5K08CA230172. https://acknowledgments.alliancefound.org. The funders had no role in study design, data collection and analysis, decision to publish, or preparation of the manuscript.

**Competing interests:** I have read the journal's policy and the authors of this manuscript have the following competing interests: WAW reports that he is an advisor and has received equity from Koneksa Health, and that he is on the strategic advisory board for the Digital Medicine Society. EB reports that he has been an advisor for Sivan, Navigating Cancer, Carevive, and AstraZeneca. The authors otherwise declare no Competing Financial or Non-Financial Interests.

reported symptom burden emerged as strong predictors of physical function (marginal $R^2$ 0.429–0.433, conditional $R^2$ 0.816–0.822).

   **Trial Registration: Clinicaltrials.gov Id** NCT02786628.

## Author summary

Performance status acquisition relies on clinician judgment though additional data sources could inform its assessment. Physical performance testing is safe in patients with cancer undergoing treatment, though the feasibility of obtaining cardiopulmonary exercise testing during routine care is unclear. Patient-generated health data acquisition during cancer treatment is feasible but the contribution of these data to understanding performance status is not known. In this multicenter observational study, we used fitness trackers in addition to validated survey instruments as a means of remotely and continuously monitoring patient physical function, a concept closely related to performance status. We found that this approach was more feasible than advanced physical performance testing during routine cancer care. Daily physical activity, heart rate, and patient-reported symptom burden provided meaningful information relevant to physical function. Prospective studies analyzing these data in the context of clinical endpoints are needed to determine whether this type of assessment could be used in place of traditional performance status assessment. Multicenter consortia could facilitate development of refined models in cancer patients and identify opportunities for interventions to improve clinical outcomes.

## Introduction

Understanding physical function during cancer treatment is critically important to patients, clinicians, regulators, and researchers to help inform treatment decision-making and to assess the tolerability of anticancer therapies. In clinical oncology, the concept used to represent this information is "performance status," which is usually assigned by treating clinicians during an office visit based on one of two commonly used scales: the Eastern Cooperative Oncology Group performance status (ECOG PS), [1] or the Karnofsky Performance Status (KPS).[2] Performance status helps to predict the benefit and tolerability of cancer treatment and helps to prognosticate long-term clinical outcomes.

   Ascertaining performance status is fraught with considerable limitations including subjective assessment by clinicians, lack of patient self-reported status, and in-office evaluation that may not fully reflect the patient experience occurring outside the healthcare setting. There is emerging interest in augmenting or optimizing routine performance status assessment to enhance its validity. Potential modalities include patient-reported questionnaires (e.g., a patient-reported performance status measure,[3] or a physical function questionnaire[4]), physical performance testing (e.g., cardiopulmonary exercise testing,[5] six-minute walk distance, [6] short physical performance battery[7], chair stands), or use of mobile health devices and sensors to acquire information from home-based settings.

   As wearable health sensors for biometric monitoring have become more ubiquitous, there has also been interest in determining whether data from these devices could also inform performance status assessment.[8] Sensor-derived data can be collected passively and continuously, with the potential to add rich detail that reflects the entirety of the lived experience of

patients. The utility of sensor-derived data have been explored in several areas, including in cardiology [9] and oncology,[10,11] though integrating these data with patient-reported data may be needed to best represent clinical concepts.

The feasibility and relevance of potential approaches to inform performance status assessment in routine cancer care is unclear. In this Alliance for Clinical Trials in Oncology pilot study (A19_Pilot2), we enrolled cohorts of patients undergoing intensive cancer chemotherapy and acquired physical testing-derived, patient-reported, and sensor-generated data[12] for each participant over 6 weeks of observation. Our patient-reported standard in this study was patient-reported physical function, a concept closely related to performance status. Our goals in conducting the study activities were to examine the feasibility of a multi-modality approach to performance status assessment and to propose candidates for patient-generated performance status assessment for future study.

## Methods

### Study design

This was a multi-institutional single-arm observational study in patients receiving chemotherapy for solid tumor malignancies or hematologic malignancies, or in patients who had planned hematopoietic cell transplantation. Baseline physical performance testing occurred after enrollment, and weekly patient-reported survey data and continuous sensor data were collected over the ensuing six weeks. The primary endpoint of the study was defined as feasibility of 4 metrics: 1) proportion of patients agreeing to participate in the study, 2) proportion of patients completing CPET and 6MWT without testing-related adverse events, 3) proportion of recruited patients wearing a FitBit at least 8 hours per day, and 4) proportion of patients for whom all patient generated data is recorded in the appropriate database. The cutoff for feasibility was set at 75%.

### Setting

Six member sites in the Alliance for Clinical Trials in Oncology were offered and agreed to participation in this multicenter study. Two sites were unable to complete physical performance testing requirements and/or had other barriers to activation, and the final multicenter cohort comprised of four Alliance sites (University of North Carolina, Dana Farber Cancer Institute, University of Chicago, Ohio State). The methods were performed in accordance with relevant guidelines and regulations and approved by the University of North Carolina Institutional Review Board as well as the institutional review board at each participating site. Written informed consent was provided by all participants. The study was registered on clinicaltrials. gov on 6/1/2016 (NCT02786628).

### Participants

**Original cohort.** Patients over the age of 18 years of age undergoing active cancer treatment were eligible and were enrolled into one of three cohorts: solid tumor malignancies, hematologic malignancies (non-transplant), or pre-hematopoietic cell transplant. Data capture was anchored around the start of a chemotherapy cycle (not required to be first cycle of therapy) or transplant. Patients were assessed weekly during the 6-week study period. Self-reported physical function was captured at each assessment using the PROMIS Physical Function 20 item questionnaire (20a) and symptom burden was captured using the Patient-Reported Symptom Monitoring System (PRSM), an instrument designed to be very similar to the PRO-CTCAE before the PRO-CTCAE was available for general use. [13,14] Aerobic capacity

was assessed at baseline using the 6-minute walk test (6MWT) and Cardio Pulmonary Exercise Testing (CPET).[6,15] For the remainder of the study, patients were given Fitbit Charge HR fitness trackers to wear for the duration of the observation period.

## Data sources/controls

The CPET and 6MWT were administered according to established protocols. Patient-reported instruments were administered electronically through the University of North Carolina PRO Core survey administration system. The wearable sensor used was the Fitbit Charge HR, which monitors continuous heart rate and physical activity data along with daily metrics summarizing activity, heart rate, and sleep. Per the device manufacturer, continuous data are deleted by the device after 7 days if they are not synchronized to conserve storage and daily aggregates are deleted after 30 days. Fitbit data were aggregated by API into Fitabase (Fitabase, San Diego, CA), a central data storage platform for Fitbit devices.

## Variables

PROMIS Physical Function was chosen due to its comprehensive and granular assessment of physical function. For the PROMIS Physical Function 20a instrument, the summary T-score is a standardized measure with a population average score of 50 and standard deviation of 10.[16,17]

Symptom burden was assessed using the Patient Reported Symptom Measure (PRSM). This instrument contains 16 individual symptom questions graded with a 1–5 severity rating, which are summed to provide an aggregate symptom burden score.[18]

Sensor-derived measures used in the analysis included Fitbit-provided statistics for activity, heart rate and sleep. Additional sensor-derived composite variables included a median heart rate calculated on a weekly basis, and a sensor-derived gait speed based on the median daily total distance and activity level (very active, moderately active, and lightly active).

A continuous sensor-derived 6MWT was calculated for each patient for every six-minute interval with high resolution data (S1 Table). Time periods in which high resolution data were not available were excluded. Sensor-derived 6MWT data were summarized as the average of the daily maximum derived distances per one-week period.

A baseline sensor-derived 6MWT was calculated based on high resolution sensor data for the first week that the participant had non-missing data. Most participants (N = 26/44, 59.0%) had sensor data available within the first three weeks after enrollment, while others (N = 10/44, 22.7%) had data available starting from the fourth week onward, usually related to the timing of the start of chemotherapy and baseline assessments. There were 8 participants (18.2%) who did not have usable sensor data.

## Statistical methods

Results analyzed were available in our database as of January 1, 2022. Linear regression analysis [19] was used to develop a model predicting the baseline Physical Function T-score using sensor-derived 6MWT and aggregate symptom burden.

Mixed effects linear regression with a subject-specific random intercept were used to predict the Physical Function T-Score using a combination of continuous sensor-derived and weekly patient-reported variables. Tracker-derived sleep data was compared to self-reported sleep symptoms using ANOVA testing. All statistical analyses and derived variable calculations were performed using a custom set of scripts written in the R programming language (version 4.1.0). Commonly used functions were organized into an R package. Tracker-derived sleep data were compared to self-reported sleep symptoms using ANOVA testing.

To account for tracker data completion and determine its association with synchronization, both sensor synchronization counts and days between sensor synchronization were compared between days with missing and non-missing data using a t-test. Missing days were categorized based on availability of heart-rate data and aggregate data for each day. Data completeness for heart rate and activity on a per day, per participant basis was labeled and tabulated for the duration of study. Hierarchical clustering was then performed based on the assigned data completeness label to categorize participants.

To identify-patient reported outcome measures eliciting similar responses in the physical function and symptom burden instruments, unsupervised analyses were conducted using hierarchical clustering performed using Euclidean distance and Ward's method. Clusters were identified by cutting the dendrogram at a distance that maximized in-cluster similarity. These groups were then considered for inclusion as variables in the models above.

## Results

### Data collection

Fifty-four patients were screened and provided written informed consent for the study between 3/30/2016 and 11/30/2017 (see CONSORT diagram, Fig 1). Ten patients subsequently withdrew following consent and before completing study activities, either because they were too ill to proceed (N = 2) or for unspecified reasons (N = 8). The remaining 44 patients (81% of those screened) comprised the study cohort, with study and patient characteristics shown in Table 1. Two patients left the study early but were able to contribute baseline testing, survey, and tracker data. Sensor data collection for these participants are shown in Fig 2. The intent of the study was to anchor baseline testing and weekly PRO collection around the start of a cycle of chemotherapy or transplant; challenges related to scheduling the CPET and timing the chemotherapy cycle meant that the amount of time for sensor data collection varied across participants. Additionally, some participants opted to continue wearing the sensor and contribute data following the last survey administration. Among the study cohort, a total of 30/44 patients (68%) were able to schedule and complete both a CPET and 6MWT assessment at baseline. Thirty-seven (84%) had usable tracker recorded step data, defined as at least 100 steps per day for at least two days per week and 34 (77%) had usable daily median heart rate data. Forty-one (93%) completed baseline patient-reported surveys while 34 (77%) completed at least 2 weekly follow-up sets of surveys. Out of 235 follow-up survey assessments that could have been completed by participants contributing baseline data, 155 (66%) were completed.

### Hierarchical clustering reveals distinct groups of participants in the domains of physical function and symptom burden

To better understand characteristics of patient-reported physical function in our study cohort, we analyzed individual PROMIS Physical Function survey items. Hierarchical clustering revealed three distinct question item groups that differentiated two groups of patients with respect to self-reported physical function (our representation of performance status). The item groups could be characterized as those assessing the ability of the participant to perform light, moderate, or strenuous physical activities. Answers on the 1–5 scale for these items showed that the items assessing ability to perform moderate or strenuous activities were those that best differentiated the two groups of participants (Fig 3). We also applied hierarchical clustering to responses to the individual symptom severity questions, which revealed two groups of participants with higher vs lower overall symptom burden (Fig 4).

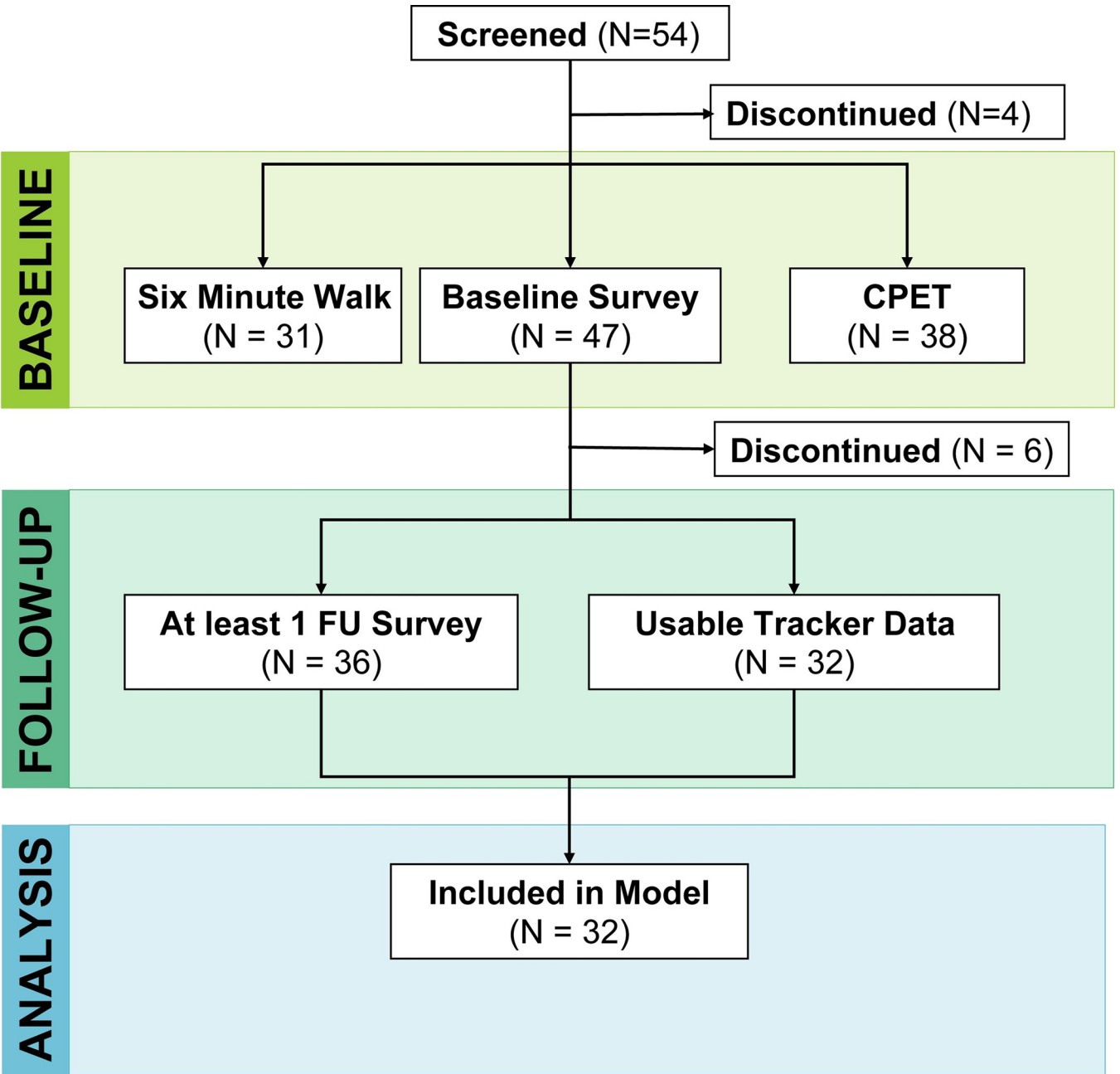

**Fig 1. Study design and participant enrollment.** Participants were initially screened and underwent baseline assessments (Baseline Survey, Six Minute Walk Test, Cardiopulmonary Exercise Testing) followed by fitness tracker data and follow-up surveys. 4 participants discontinued before baseline testing and 6 participants discontinued after completing baseline surveys. A total of 32 participants had survey data with tracker data for the corresponding time period and were included in the mixed-methods regression models.

### Data completeness is greater with increased device synchronization frequency

We examined data completeness as a function of synchronization between the device and the Fitbit servers; synchronization occurred every 23 days on average (SD 28) and frequency of synchronization was associated with missing data (p<0.001) while the total number of

**Table 1. Study Cohort Descriptive Data (N = 44).**

|  | N | % |
|---|---|---|
| Age | 53 +/- 15 |  |
| Female | 29 | 66% |
| Male | 15 | 34% |
| Race |  |  |
| White | 32 | 72% |
| Black | 5 | 11% |
| Asian | 2 | 5% |
| Other | 3 | 7% |
| Unknown | 2 | 5% |
| Hematopoietic Cell Transplantation |  |  |
| Myeloablative conditioning | 14 | 32% |
| Non-myeloablative or uncategorized conditioning | 7 | 16% |
| Solid Tumor Malignancy |  |  |
| Breast | 9 | 20% |
| Colorectal | 2 | 5% |
| Pancreas or gallbladder | 4 | 9% |
| Hematologic Malignancy (non-transplant) |  |  |
| Lymphoma or chronic lymphocytic leukemia | 5 | 11% |
| Multiple Myeloma | 3 | 7% |

synchronizations was not. Most data were synchronized within 30 days (81.87%). Aggregate data summarizing total activity were more complete than continuous heart rate and activity data. Based on data completeness, hierarchical clustering revealed four distinct groups of individuals: participants with 1) complete data capture, 2) complete data capture at the beginning of the study, 3) complete data capture toward the end of the study and 4) mostly incomplete data (S1 Fig). The number of patients in each category was roughly similar, suggesting that future studies will need a multimodal approach to ensuring data completeness throughout the study period. Trends in patient-reported and sensor-derived data in the study cohort over time are shown in S2, S3 and S4 Figs.

## Physical performance testing can be used to derive a sensor variable which predicts physical function

We analyzed baseline physical performance testing and compared our sensor-derived 6MWT with the in-person 6MWT (S5 Fig). Only participants with high resolution sensor data on the date of the 6MWT were included. For these 15 participants, the sensor-derived 6MWT and in-person 6MWT were significantly associated ($R^2$ 0.21, $\rho = 0.52$, $p<0.05$), with a one-meter increase in in-person 6MWT associated with a 0.74-meter increase in the sensor-derived 6MWT. Next, we explored the association of the baseline sensor-derived 6MWT to baseline patient-reported physical function. Using a linear model that adjusted for baseline symptom burden in 36 (82%) patients with available data, we found that a 10-meter increase in the sensor-derived 6MWT was associated with a 0.3 unit increase in self-reported physical function T score ($p<0.01$). Higher baseline symptom burden was associated with poorer baseline physical function ($\beta = -0.6$, $p<0.01$). However, sensor derived 6MWT resulted in low interclass correlation coefficients (0.32–0.58) indicating high variability in the metric. This likely resulted from sparse high resolution data storage due to infrequent tracker synchronization. As a result, it was not incorporated in subsequent models.

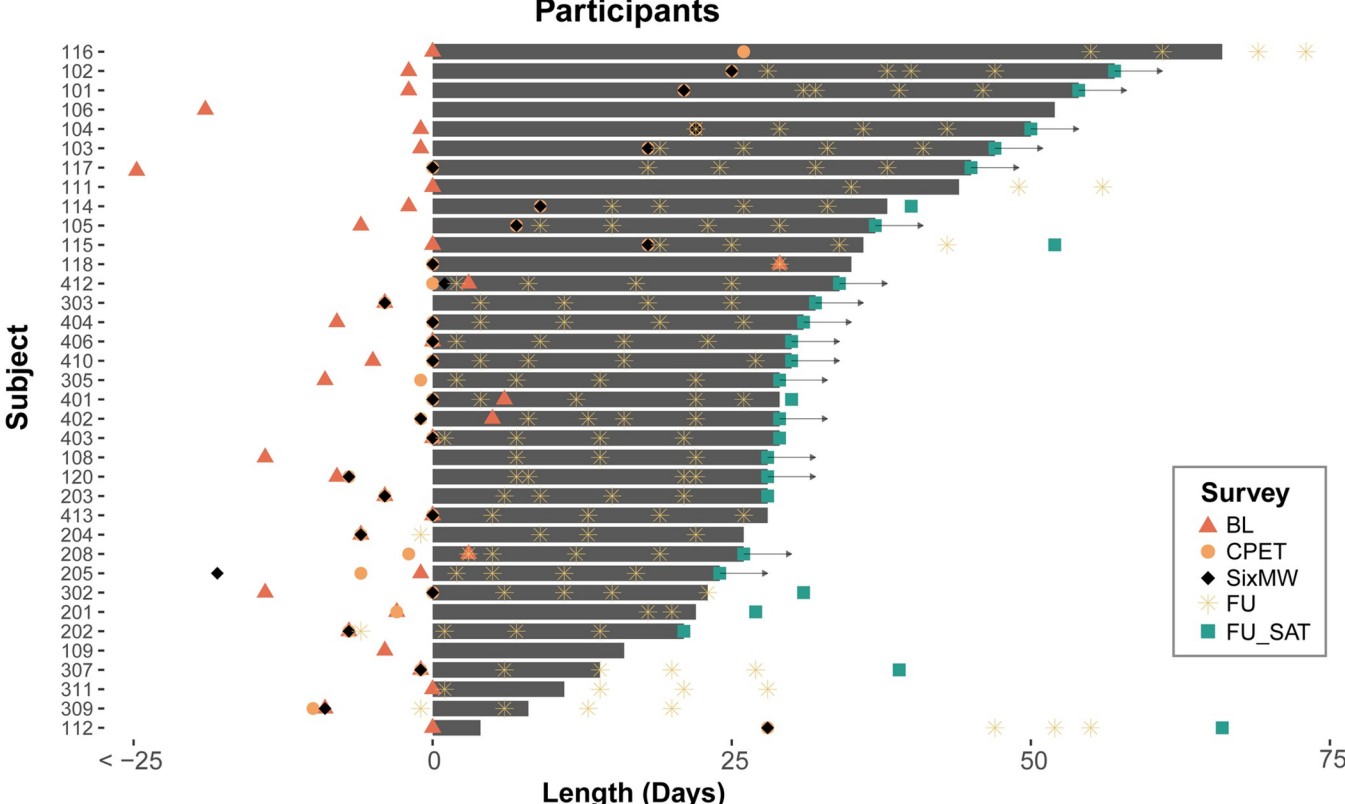

**Fig 2. Swimmer's plot of tracker data collection relative to survey instrument administration and baseline exercise physiology testing.** Participants who continued wearing trackers following formal completion of the study period are marked with arrows (BL: Baseline Survey, CPET: Cardiopulmonary Exercise Testing, SixMW: Six Minute Walk Test, FU: Follow Up Survey, FU_SAT: Follow Up Satisfaction Survey). Delays from enrollment to acquisition of sensor data were generally related to timing of the start of chemotherapy and baseline assessments.

### Sensor-derived sleep data are associated with self-reported insomnia scores

Because insomnia was a contributor to symptom burden differentiation, we explored the relationship of sensor-derived sleep data to self-reported insomnia severity on the PRSM questionnaire. We found that the sensor-derived sleep variables most strongly associated with self-reported insomnia severity were sleep efficiency, number of restless episodes during sleep, and restless duration during sleep ($p < 0.05$ for each, S2 Table).

### Average daily activity, heart rate, and weekly symptom burden predict weekly physical function

We constructed regression models with repeated measures that used daily summary metrics for sensor data averaged over a week, and weekly patient-reported data (Table 2, S6 Fig). With this approach, 32 (73%) patients with corresponding survey and sensor data contributed to the model. The outcome of the model was patient-reported physical function, and potential contributors were sensor-derived variables and symptom burden. In one model (Model A), we used variables which should be obtainable from most sensors (median heart rate, daily steps as measured by activity counts), and in the other model (Model B), we utilized Fitbit-provided variables (Light Active Minutes, Very Active Minutes) that were calculated by Fitbit-specific proprietary algorithms. Both models performed similarly to predict physical function

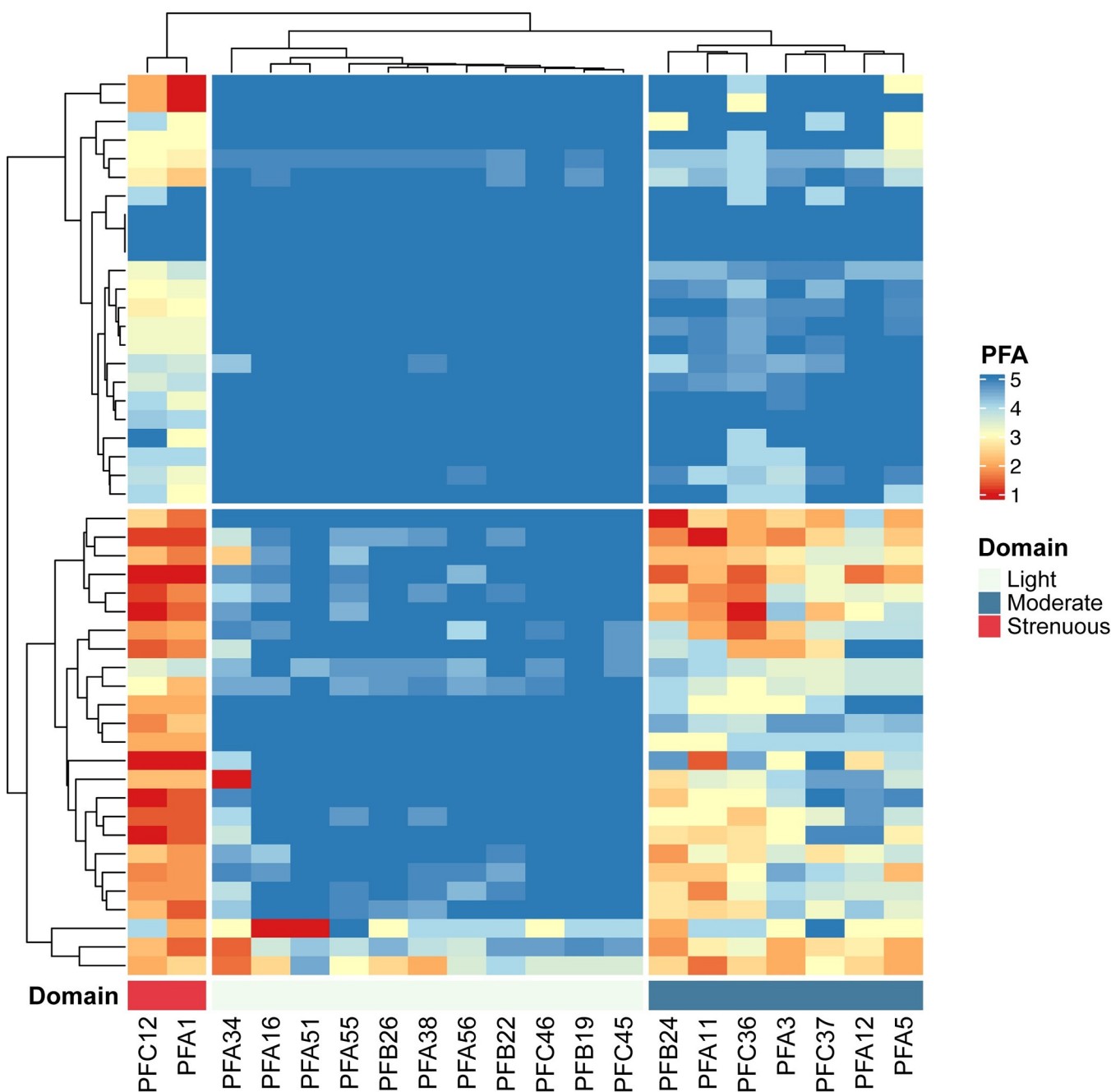

**Fig 3. Hierarchical clustering of average survey question results for each patient.** This unsupervised analysis revealed distinct domains of activity within the Promis Physical Function 20 question instrument for the cohort and that differences between patients were largely driven by questions assessing strenuous and moderate physical activity.

(marginal $R^2$ 0.429–0.433, conditional $R^2$ 0.816–0.822) and demonstrated that heart rate, symptom burden and activity measures were all significant contributors. The mean patient age in the modeled group was 55.3 years (43.2 years among those not included), and there was no significant difference in demographics among those in or excluded from the model (Fisher's exact p = 0.3).

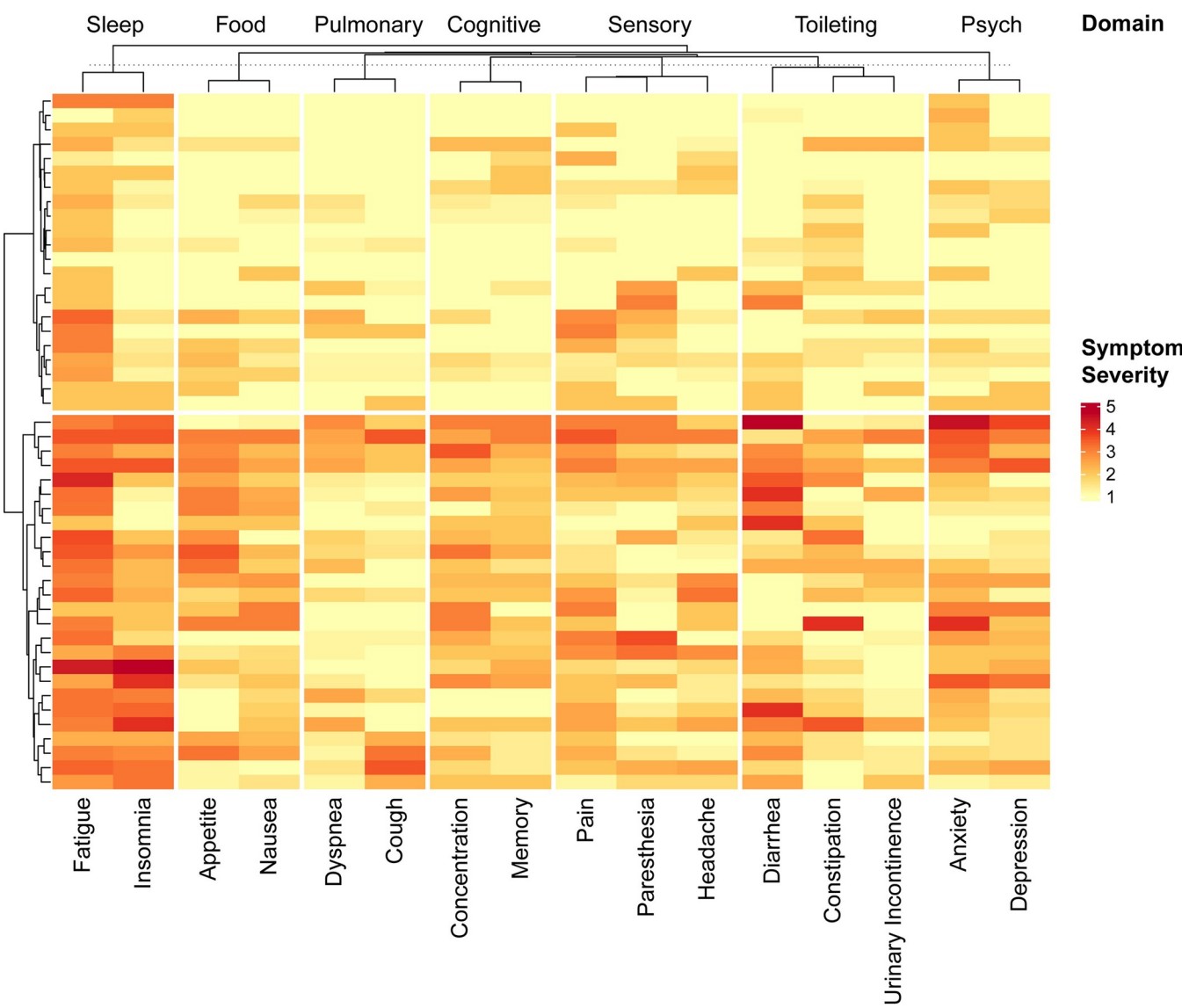

**Fig 4. Symptom burden in the PRSM symptom burden instrument.** Hierarchical clustering of average responses in the original cohort revealed two distinct cohorts of patients with higher and lower symptom burdens. Questions were organized into clinical domains based on the individual questions and respective symptoms.

We also fit mixed-methods models based on individual symptom burden questions along with clinical domains based on item similarity. After pruning, questions reporting fatigue, impaired memory, anxiety, and nausea were the most significant contributors to the model (S3 Table). These questions showed similar contributions to both the generic model (Model A) and the Fitbit variable model (Model B).

## Discussion

In this pilot study, we found that patient-generated health data were feasible to obtain during routine cancer treatment and provided meaningful information about patient-reported physical function, a concept closely related to performance status. We believe that this overall

**Table 2. Physical function T-score prediction using sensor data and symptom burden.** The generic variable model (left) includes variables that should be available from most tracker models while the Fitbit-specific variable model (right) uses Fitbit-specific variables derived from proprietary algorithms. Marginal $R^2$ values, which consider non-patient specific (or fixed) effects, are similar in both models, indicating similar goodness of fit. Conditional $R^2$ values, which incorporate both patient specific and fixed effects are also similar.

| | Model A: Generic variable model | | | | Model B: Fitbit variable model | | | | |
|---|---|---|---|---|---|---|---|---|---|
| *Predictors* | (Intercept) | Median HR | Total Steps | Symptom Burden | (Intercept) | Median HR | Light Active Minutes | Very Active Minutes | Symptom Burden |
| *Estimates* | 46.61 | -1.55 | 2.13 | -3.24 | 46.59 | -1.61 | 1.54 | 0.79 | -3.29 |
| *CI* | 44.91–48.32 | -2.72 –-0.37 | 0.83–3.43 | -4.40 –-2.09 | 44.89–48.30 | -2.80 –-0.41 | 0.13–2.95 | 0.02–1.56 | -4.45 –-2.13 |
| *p-value* | <0.001 | 0.01 | 0.001 | <0.001 | <0.001 | 0.008 | 0.032 | 0.045 | <0.001 |
| **Marginal R2** | 0.433 | | | | 0.429 | | | | |
| **Conditional R²** | 0.822 | | | | 0.816 | | | | |

message should lead to future work refining measurement approaches and developing interventions to improve performance status over time.

We encountered significant obstacles to implementing physical performance testing, particularly cardiopulmonary exercise testing, in our routine care design. Previous studies have shown that cardiopulmonary exercise testing is safe in patients undergoing cancer treatment. [5] However, we found that some centers were unable to identify facilities to perform this testing, and in those centers which did participate, scheduling logistics, patient willingness to undergo CPET, and timing CPET around chemotherapy were challenges. Study staff at some centers were unable to coordinate concomitant 6-minute walk testing with CPET, leading to significant missing data. While CPET and/or 6MWT could be useful in specific scenarios where treatment is planned in advance, like surgery[20] or transplant,[21] or for baseline evaluations for clinical trials, we did not find it feasible to implement testing during routine cancer care. Others have reported challenges with implementing less intensive physical performance tests than CPET because of constraints related to cancer clinic space or personnel. [22] This is consistent with general clinical practice in which physical performance testing is uncommon. For those who wish to use measures other than CPET, we recommend exploring potentially less clinically disruptive implementation approaches, such as virtual administration or coordination within specialized consultative clinics.

Although acquiring patient-generated data from surveys and sensors was easier to implement than CPET and 6MWT, we nevertheless encountered incomplete data. Our patient-reported survey completion rates were lower than in other clinical trial settings.[23] We also identified unique groups of patients with varying levels of Fitbit wear, with some having low level of adherence. We found that the frequency of sensor syncing limited the availability of high-resolution sensor data, and some participants did not sync frequently. For these reasons, we recommend that others who implement patient-generated health data collection consider best practices and novel strategies to maximize adherence.[24] These could include making data available back to participants; automated reminders to complete surveys or wear the sensor; phone call "boosters" to motivate continued participation; incorporation within an intervention where participants feel that their data will be used to improve their care; placement of automated synchronization devices in clinician offices; and incentivizing higher frequency syncing.

We chose the patient reported PROMIS Physical Function instrument as our measure of physical function. We did this because clinical performance status by ECOG or KPS is a measure of patient functioning, and a validated physical function instrument that is completed by

patients captures similar information. However, we acknowledge the absence of consensus on the use of the PROMIS physical function measure as a stand-in for performance status. Several alternative measures exist, some longer than others, that could also be considered–from a single item patient-reported performance status measure to inventories of activities of daily living to subscales of other validated quality of life measures (e.g., EORTC QLQ-C30, SF-36). An optimal performance status measure should be meaningful to patients and clinically useful. Further, current clinical performance status measurement may incorporate subconscious clinician judgment based on other features that are prognostically relevant but not directly related to patient functioning (e.g., disease biology or progression), and these factors may contribute to the current utility of clinical measurement. An optimized patient-reported performance status measure will not account for these clinician biases, which will need to be incorporated into medical decision making in other ways. We acknowledge that the absence of clinician-rated ECOG and KPS in our data set was a limitation in our analysis, and we recommend inclusion of longitudinal clinician-reported ECOG and KPS in future studies in order to better sort out the contributions of various potential measures of performance status.

We found that several measures obtained from patient-generated data–symptom burden, physical activity, and heart rate–were associated with performance status. These data could be feasibly acquired from patients in the home setting using an electronic survey platform and an activity tracker. Because generic data (e.g., total daily steps, median heart rate) could be used in the statistical models, it is likely that other types of sensors could be used in similar calculations with minimal modifications. In addition, we found that the increased quantity of sensor and survey data provided by repeated longitudinal assessments led to better predictions, as seen in our repeated measures analysis. This suggests a role for continuous, longer duration monitoring in future studies. We acknowledge that, to date, data from wearable activity monitors have demonstrated a weak to moderate association with clinician performance status, as reviewed by Kos et al., though previous studies have primarily focused on data related to steps and activity intensity.[8]

From a measurement perspective, we acknowledge that our work represents early steps towards an appropriately developed and validated approach for a PGHD-based performance status measurement model. In considering rigorous digital measure development, we recognize recent standards-setting work in the field, such as the V3 model (verification, analytical validation, clinical validation) as applied to Biometric Monitoring Technologies (BioMeTs). [25] Additionally, widespread use of sensors will require attention to issues around data governance and cybersecurity, and an understanding of the data lifecycle from participant to sensor to cloud and beyond.[26,27] Because of considerations related to patient sample size, the vast proliferation of different types of commercial and health-grade sensors, and the many nuances of study design and analysis in this area, it is unlikely that this work will be accomplished by single centers. We believe that multicenter projects are needed and can be facilitated under the standards setting and convening power of multiple groups including cooperative clinical trials groups such as the Alliance, professional societies such as the American Society of Hematology (ASH) and the American Society of Clinical Oncology (ASCO), the US Food and Drug Administration, and the National Institutes of Health.[28] Multicenter real world data platforms maintained by these groups should accommodate development and implementation of patient-generated health data measures for research and clinical use. Consideration could also be given toward development of "oncology digital centers of excellence" that could be activated for multiple measurement projects under different convening mechanisms.

It is biologically plausible that symptom burden, physical activity and heart rate could be meaningful inputs to performance status. Symptom burden may reflect treatment tolerability, [29] and the cumulative impact of disease and treatment upon performance may be as

important as any one individual severe symptom. Physical activity reflects behavioral characteristics, the impact of disease and treatment, and underlying aerobic capacity. Heart rate may reflect underlying fitness, the effects of treatment, and relevant autonomic inputs.[30] Multimodal interventions that include exercise prescription, supportive care and navigation may be appealing for future studies that seek to maintain or improve performance over time. Other types of interventions that could be considered include real-time monitoring and decision-making on clinical trials or in advance of high intensity therapies such as leukemia induction chemotherapy or hematopoietic cell transplantation.

In conclusion, we found that although CPET measurement was not feasible for incorporation into routine cancer care, sensor-derived and patient-reported data provided meaningful information relevant to longitudinal performance status assessment. We identified limitations and lessons learned from our measurement approaches. We believe that this work represents a proof-of-concept analysis that can be taken forward to future studies to better understand and improve performance status during cancer treatment.

## Supporting information

**S1 STROBE Checklist. STROBE Statement—checklist of items that should be included in reports of observational studies.**
(DOCX)

**S1 Fig. Tracker data completeness. Hierarchical clustering revealed four distinct groups of participants, based on completeness of activity and heart rate data across available potential days wearing the sensor.**
(TIF)

**S2 Fig.  S2a Fig. Longitudinal Physical Function T-score assessments (original cohort).** Longitudinal analysis of T-Score vs days since baseline survey completion. Hierarchical clustering revealed two distinct groups of patients, with low and high PROMIS Physical Function outcomes, designated by color schemes. Some participants maintained consistent results while others deviated between weeks. Only participants with data for all time periods were included in this figure. Higher T-Scores are associated with improved performance. **S2b Fig. Longitudinal symptom burden (original cohort).** Hierarchical clustering of symptom burden revealed two participant cohorts with low and high symptom burdens, as per the respective blue and red color schemes. The spaghetti plot below depicts total symptom burden scores in relation to days since baseline survey completion. Only participants with PRSM sums for every week were included. Higher PRSM sums are associated with increased symptoms.
(TIF)

**S3 Fig. Aggregate distribution of Physical Function T-score, symptom burden, and sensor-derived 6MWT.**
(TIF)

**S4 Fig. Distribution of Physical Function T-score, symptom burden, sensor-derived 6MWT, median heart rate, and total activity.**
(TIF)

**S5 Fig. Correlation matrix containing symptom burden, T-Score, VO2 max, clinical six-minute walk distance, and derived six-minute walk from tracker data.**
(TIF)

**S6 Fig. Correlation matrix containing symptom burden, T-Score, and fitness tracker variables used in mixed-methods linear regression models.**
(TIF)

**S1 Table. Variables used in the analysis.**
(XLSX)

**S2 Table. ANOVA modeling for relationships between sensor-derived sleep data and self-reported insomnia severity.**
(XLSX)

**S3 Table. The generic and Fitbit variable models with symptom burden predictors: both as clinical domains and individual questions.** Models were subsequently pruned to only include predictive symptom burden variables.
(XLSX)

## Acknowledgments

The content is solely the responsibility of the authors and does not necessarily represent the official views of the National Institutes of Health.

## Author Contributions

**Conceptualization:** William A. Wood, Natalie S. Grover, Jacob Glass.

**Data curation:** William A. Wood, Jacob Glass.

**Formal analysis:** William A. Wood, Deepika Dilip, Andriy Derkach, Jacob Glass.

**Funding acquisition:** William A. Wood.

**Investigation:** William A. Wood, Jacob Glass.

**Methodology:** William A. Wood, Deepika Dilip, Andriy Derkach, Jacob Glass.

**Project administration:** William A. Wood, Charlotte Bailey.

**Supervision:** Olivier Elemento, Jacob Glass.

**Validation:** Deepika Dilip, Andriy Derkach, Jacob Glass.

**Visualization:** Deepika Dilip, Andriy Derkach, Jacob Glass.

**Writing – original draft:** William A. Wood.

**Writing – review & editing:** William A. Wood, Deepika Dilip, Natalie S. Grover, Olivier Elemento, Ross Levine, Gita Thanarajasingam, John A. Batsis, Charlotte Bailey, Arun Kannappan, Steven M. Devine, Andrew S. Artz, Jennifer A. Ligibel, Ethan Basch, Erin Kent, Jacob Glass.

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
