## [Decision Letter · Decision Letter 0]

3 Aug 2022

PDIG-D-22-00147

Wearable sensor-based performance status assessment in cancer: a pilot multicenter study from the Alliance for Clinical Trials in Oncology (A19_Pilot2)

PLOS Digital Health

Dear Dr. Wood,

Thank you for submitting your manuscript to PLOS Digital Health. After careful consideration, we feel that it has merit but does not fully meet PLOS Digital Health's publication criteria as it currently stands. Therefore, we invite you to submit a revised version of the manuscript that addresses the points raised during the review process.

The editor's and reviewers' comments are attached below. Please note the significant amount of work that is required to address the concerns, for example about novelty, before the manuscript can be further considered.

Please submit your revised manuscript within 60 days Oct 02 2022 11:59PM. If you will need more time than this to complete your revisions, please reply to this message or contact the journal office at digitalhealth@plos.org. Please include the following items when submitting your revised manuscript:

We look forward to receiving your revised manuscript.

Kind regards,

Sung Won Choi

Guest Editor

PLOS Digital Health

Journal Requirements:

1. Please amend your detailed online Financial Disclosure statement. This is published with the article. It must therefore be completed in full sentences and contain the exact wording you wish to be published.

State what role the funders took in the study. If the funders had no role in your study, please state: “The funders had no role in study design, data collection and analysis, decision to publish, or preparation of the manuscript.”

2. Please update the completed online 'Competing Interests' statement. Please declare all competing interests beginning with the statement “I have read the journal's policy and the authors of this manuscript have the following competing interests:”.

3. In the online submission form, you indicated that “Data supporting the results reported in the article can be found upon request by contacting the corresponding author.”. All PLOS journals now require all data underlying the findings described in their manuscript to be freely available to other researchers, either 1. In a public repository, 2. Within the manuscript itself, or 3. Uploaded as supplementary information.

4. Please provide separate figure files in .tif or .eps format and remove any figures embedded in your manuscript file. Please also ensure that all files are under our size limit of 10MB.

For more information about how to convert your figure files please see our guidelines: https://journals.plos.org/digitalhealth/s/figures

5. We notice that your supplementary figures and tables are included in the manuscript file. Please remove them and upload them with the file type 'Supporting Information'. Please ensure that each Supporting Information file has a legend listed in the manuscript after the references list.

Additional Editor Comments (if provided):

While finding merit in the authors’ work, the manuscript and associated reviewers’ comments are not convincing that the manuscript meets the priority needed for publication at this time. In addition to the reviewers' comments, the major concern relates to the (lack of) novelty of the manuscript and how it advances the field, primarily due to the stated nature of feasibility and acceptability (poorly defined). Given that a number of other studies have demonstrated the feasibility of collecting wearable device data during routine cancer treatment and significant associations with performance status, please address the innovation or importance of this work to the field.

Reviewers' comments:

Reviewer's Responses to Questions

**Comments to the Author**

1. Does this manuscript meet PLOS Digital Health’s publication criteria? Is the manuscript technically sound, and do the data support the conclusions? The manuscript must describe methodologically and ethically rigorous research with conclusions that are appropriately drawn based on the data presented.

Reviewer #1: Yes

Reviewer #2: Yes

2. Has the statistical analysis been performed appropriately and rigorously?

Reviewer #1: I don't know

Reviewer #2: Yes

3. Have the authors made all data underlying the findings in their manuscript fully available (please refer to the Data Availability Statement at the start of the manuscript PDF file)?

Reviewer #1: No

Reviewer #2: Yes

4. Is the manuscript presented in an intelligible fashion and written in standard English?

Reviewer #1: Yes

Reviewer #2: Yes

5. Review Comments to the Author

Reviewer #1: This study aims to address important limitations of existing measures of performance status, and strengths include the longitudinal design and inclusion of objective, patient-reported, and sensor-derived measures of performance.

Major Concerns

• The methods and results sections feel disorganized and out of sync with each other and with the stated and preregistered aims of the study. If the primary objective was to assess the feasibility of this measurement approach, feasibility should be better defined, and analyses that are not central to the aims of this work omitted. For example, what is the purpose of comparing baseline CPET and baseline 6MWT? Since validation of sensor metrics was not named as a goal, why compare tracker-derived sleep data to self-reported sleep data? What was the purpose of the hierarchical clustering of individual items/Figures 3 and 4?

• Given the introduction’s initial focus on clinician-assigned performance status, I expected the paper to consider how patient-generated data related to a more traditional clinician-based measure. Was clinician-rated ECOG PS or KPS assessed? How does ECOG or KPS correlate with PROMIS Physical Function in this or previous studies?

• Given that a number of other studies (reviewed in Kos et al., 2021) have demonstrated the feasibility of collecting wearable device data during routine cancer treatment and significant associations with performance status, please clarify what is the novel contribution of this manuscript to the field.

Minor Concerns

• How did the 32 patients included in the models differ from the 12 enrolled and included in the Table 1 study cohort but who did not have sufficient data available with regard to demographics?

• Relatedly, any significant differences in feasibility of patient-generated data collection across the three different cohorts of patients?

• Which result does the following statement from the Discussion refer to: “Increased amounts of data, from both surveys and sensors, led to better predictions, as seen in our repeated measures analysis, suggesting a role for continuous, longer duration monitoring.”

• Why was data synchronized to the Fitbit servers only every 23 days on average? Were patients instructed to sync the data at regular intervals? Also the association between synchronization frequency and amount of missing data seems obvious.

• While not a stated goal of the study and based on a tiny subsample, the comparison between in-person and sensor-derived 6MWT is interesting, especially given the challenges noted in obtaining in-person 6MWT.

Reviewer #2: The manuscript by Wood et al is a prospective observational feasibility study examining the use of objective data sources and structured patient generated health data tools to assess clinical performance status during routine clinical care done through the alliance. The project is in an area of high need which is to help improve objective assessments of performance status while leveraging wearables and validated surveys. The manuscript is well written and reflects on challenges encountered as well as optimum analysis of extensive data collected during the study. There are a few concerns:

A) This is a feasibility study but the authors don’t mention any a priori parameters to indicate feasibility. They do describe low feasibility for baseline CPET and 6MWT because it was obtained only in 68% patients but don’t mention as to what the cut-off for feasibility was.

B) My other big concern with feasibility is being able to get only 44 final patients out of 54 screened patients over a course of 20 months or so from 4 large cancer centers for the study. 

C) More details needed about the recruitment plan- were consecutive patients approached? Were the 54 screened patients a subset of a larger group of patients who were approached but declined since their inclusion criteria seems not very restrictive? How were these 54 patients selected?

D) Not clear what is meant by ‘both cohorts’ in line 202 as in the above section, the authors mention the original cohort which was divided into 3 cohorts: solid tumor vs. heme malignancy vs. pre HCT.

6. PLOS authors have the option to publish the peer review history of their article (what does this mean?). If published, this will include your full peer review and any attached files.

**Do you want your identity to be public for this peer review?** For information about this choice, including consent withdrawal, please see our Privacy Policy.

Reviewer #1: No

Reviewer #2: No

---

## [Decision Letter · Decision Letter 1]

1 Nov 2022

PDIG-D-22-00147R1

Wearable sensor-based performance status assessment in cancer: a pilot multicenter study from the Alliance for Clinical Trials in Oncology (A19_Pilot2)

PLOS Digital Health

Dear Dr. Wood,

Thank you for submitting your manuscript to PLOS Digital Health. After careful consideration, we feel that it has merit but does not fully meet PLOS Digital Health's publication criteria as it currently stands. Therefore, we invite you to submit a revised version of the manuscript that addresses the points raised during the review process.

As you can see, the manuscript was rereviewed by two reviewers. One is satisfied, while the other requests changes. 

Please submit your revised manuscript within 60 days Dec 31 2022 11:59PM. If you will need more time than this to complete your revisions, please reply to this message or contact the journal office at digitalhealth@plos.org. Please include the following items when submitting your revised manuscript:

We look forward to receiving your revised manuscript.

Kind regards,

Sung Won Choi

Guest Editor

PLOS Digital Health

Journal Requirements:

Additional Editor Comments (if provided):

Reviewers' comments:

Reviewer's Responses to Questions

**Comments to the Author**

1. If the authors have adequately addressed your comments raised in a previous round of review and you feel that this manuscript is now acceptable for publication, you may indicate that here to bypass the “Comments to the Author” section, enter your conflict of interest statement in the “Confidential to Editor” section, and submit your "Accept" recommendation.

Reviewer #1: (No Response)

Reviewer #2: All comments have been addressed

2. Does this manuscript meet PLOS Digital Health’s publication criteria? Is the manuscript technically sound, and do the data support the conclusions? The manuscript must describe methodologically and ethically rigorous research with conclusions that are appropriately drawn based on the data presented.

Reviewer #1: Yes

Reviewer #2: Yes

3. Has the statistical analysis been performed appropriately and rigorously?

Reviewer #1: Yes

Reviewer #2: N/A

4. Have the authors made all data underlying the findings in their manuscript fully available (please refer to the Data Availability Statement at the start of the manuscript PDF file)?

Reviewer #1: Yes

Reviewer #2: Yes

5. Is the manuscript presented in an intelligible fashion and written in standard English?

Reviewer #1: Yes

Reviewer #2: Yes

6. Review Comments to the Author

Reviewer #1: • Consider exchanging “physical function” for “performance status” given that patient-reported physical function was the representation of performance status selected and that more traditional clinician-rated performance status measures were not assessed.

• Please add a citation to support the claim that “exercise testing is the gold standard for performance status assessment.”

• I appreciate the authors’ edits to clarify the primary feasibility aims of this paper. How was “wearing a Fitbit at least 8 hours per day” (line 157) originally operationalized? If based on something like the availability of heart rate or step data for 8 hours per day, this actually seems more stringent than “100 steps/day for two days per week” since 100 steps can be accrued in far fewer than 8 hours and two days per week seems very minimal. What is the justification for this threshold for “usable” tracker recorded step data and the decision not to focus on 8 hours per day of Fitbit wear time?

• Line 227-228 describes analytic methods for comparing baseline CPET and 6MWT and should be omitted.

• I appreciate the authors efforts to justify the inclusion of the sleep sensor data and self-reported sleep data but still feel this is irrelevant to the paper’s focus on performance status/physical function and should be omitted (lines 234-235 & 237-238; lines 317-322 & Supplemental Table 2).

• The lack of clinician-rated ECOG and KPS should be acknowledged as a limitation of this multicenter study.

• Given relevance to this manuscript, the Kos et al 2021 review should be cited and included in the Introduction or Discussion.

• Re: line 335, please confirm that age did not significantly differ in the included participants vs. the excluded participants.

Reviewer #2: Authors have attempted to address all the comments within the constraints of the collected data. No further comments at this time

7. PLOS authors have the option to publish the peer review history of their article (what does this mean?). If published, this will include your full peer review and any attached files.

**Do you want your identity to be public for this peer review?** For information about this choice, including consent withdrawal, please see our Privacy Policy. 

Reviewer #1: No

Reviewer #2: Yes: Nandita Khera

---

## [Editor Report · Decision Letter 2]

6 Dec 2022

Wearable sensor-based performance status assessment in cancer: a pilot multicenter study from the Alliance for Clinical Trials in Oncology (A19_Pilot2)

PDIG-D-22-00147R2

Dear Dr. Wood,

We are pleased to inform you that your manuscript 'Wearable sensor-based performance status assessment in cancer: a pilot multicenter study from the Alliance for Clinical Trials in Oncology (A19_Pilot2)' has been provisionally accepted for publication in PLOS Digital Health.

Best regards,

Sung Won Choi

Guest Editor

PLOS Digital Health

The authors have adequately addressed the reviewer's queries.